# Edward A. Pace: First-Generation Psychologist, Twenty-First Century Role Model

**Keith A. Puffer** [1,*] and **Kris G. Pence** [2,*]

1   Department of Psychology, Indiana Wesleyan University, Marion, IN 46952, USA
2   Department of Counselor Education and Counseling Psychology, Western Michigan University, Kalamazoo, MI 49464, USA
*   Correspondence: keith.puffer@indwes.edu (K.A.P.); kristin.g.pence@wmich.edu (K.G.P.)

**Abstract:** In 1891, Edward A. Pace, a Catholic priest and first-generation psychologist, commenced a career at the Catholic University of America in Washington, D.C. Amidst the daunting challenges in being a professor and researcher, particularly at a newly established university, he thrust himself into a third role, apologist. Habits related to the Monsignor's three roles have contemporary relevance for psychologically-trained Protestants; in this case study, we examine four notable practices. Dr. Pace modeled an appetence for wisdom in multiple disciplines, a keen awareness of rival worldviews, intentional ripostes to Catholic critics of scientific psychology, and last, unrelenting steadfastness to the Christian faith. To characterize the priest-psychologist, we present a brief biographical sketch and an overview of influential historical movements in the zeitgeist of the late 19th and early 20th centuries affecting his life. In addition, the aforementioned habits of Pace and applications for Protestants engaging in psychology in the 21st century are delineated.

**Keywords:** case study; integrating Christianity and psychology; Edward A. Pace; first-generation psychologist; Catholicism; Protestant

## 1. Introduction

The roles that people occupy shape their lifestyle. Bronfenbrenner (1979) identified role participation (e.g., teacher, father) as an important environmental force affecting human development. In 1891, Edward A. Pace, a Catholic priest and first-generation psychologist, commenced a career at the newly established Catholic University of America in Washington, D.C. Amidst the numerous challenges in being a first-time professor and researcher, the Monsignor thrust himself into a third role, apologist.

Habits related to the priest-psychologist's three roles have contemporary relevance for psychologically-trained Protestants. In this case study, we examine four notable practices. Dr. Pace modeled an appetence for wisdom in multiple disciplines, a keen awareness of rival worldviews, intentional ripostes to Catholic critics of scientific psychology, and last, unrelenting steadfastness to the Christian faith. Yet, who was this first-generation psychologist? How do the aforementioned habits embody Bandurian traits of relevance, competency, and prestige and then merit emulation by 21st century psychological professionals who are Protestants?

The following seeks to respond to these questions. We present a brief biographical sketch on Pace and an overview of three influential historical movements in the zeitgeist of the late 19th and early 20th centuries affecting his life. Moreover, the aforementioned four habits of the Monsignor and applications for Protestants engaging in psychology in the 21st century are delineated.

## 2. Biographical Sketch

Edward A. Pace was a native Floridian, born in 1861. His parents, George and Margaret, raised him and seven siblings in the Catholic faith (Braun 1968). After high school, he pursued training for the priesthood at St. Charles College in Maryland and finished his theological/philosophical education in Rome, Italy, at the Pontifical North American College (panc.org 2019). Ordination transpired in 1885. One year later, Pace earned a doctorate in Sacred Theology at the age of 25 and agreed to return to the United States for the pastorate at the Cathedral Church of St. Augustine, Florida (Sexton 1980).

In 1888, Bishop Keane, the first rector or president of the newly established Catholic University of America in Washington, D.C., recruited Pace to become a faculty member in the philosophy department. The appointment as professor included time in Europe for additional training in science—to ensure "relevant engagement with the modern world"—particularly, with the scientific community (Kugelmann 2005, p. 133). He enrolled in a biology class at the University of Louvain (Belgium) and chemistry and physiology courses at Sorbonne (France).

From 1889 to 1891, the Monsignor studied at the birthplace of scientific psychology, the University of Leipzig in Germany. He was one of thirty-three Americans who earned a Ph.D. degree and had Wilhelm Wundt as a first or second dissertation reader. Yet, he was the only Catholic priest. According to Benjamin et al. (1992), "Americans who were drawn to Leipzig in the late 19th and early 20th centuries were there because of an enthusiasm for the prospects of the new science of the mind [and its methodology]" (p. 130). The completion of the dissertation (1891) marked the end of Pace's education in Europe. With a second doctorate at the age of 30, the priest-psychologist returned to Washington, D.C.

For the next 44 years, Dr. Pace dedicated himself to lecturing, researching, and defending scientific psychology at CUA. He retired in 1935. A three-year struggle with complications from diabetes led him to convalesce at Providence Hospital in Washington D.C. On 28 April 1938, the beloved and respected Monsignor passed at the age of seventy-seven (CUA Archives 2016).

## 3. Zeitgeist Overview

Several cultural trends and historical movements mark the zeitgeist of the late 19th and early 20th centuries. Three important historical phenomena affected Pace. The triad includes neo-scholasticism, progressivism, and empirical psychology.

### 3.1. Neo-Scholasticism

In 1879, Pope Leo XIII named neo-scholasticism or Thomism to be the ordained worldwide school of theology and philosophy in his encyclical, *Aeterni Patris*. He declared, "We exhort you to restore the golden wisdom of St. Thomas and to spread it far for the defense and beauty of the Catholic faith and for the advantage of all the sciences" (Pope 1879, p. 26). This pope reached back in time, to the 13th century, and endorsed Thomas Aquinas' ideology as a trusted system for the Catholic Church in the 19th century.

Explaining the papal decision, Del Colle (2010) noted, Leo XIII "was responding to the contemporary problem of how faith and reason needed to be integrated to enable an adequate and comprehensive response to the contestations over truth and religion" (p. 376). Concomitant with the encyclical, the pope established several institutes for training in Thomism and science. Desire-Joseph Mercier, a professor of philosophy and psychology at the University of Louvain in Belgium, developed an influential institute in 1889 and created an experimental psychology laboratory two years later (Kugelmann 2005).

Edward A. Pace was an undergraduate at St. Charles College in Maryland when *Aeterni Patris* was promulgated. Subsequently, he was trained in scholastic theology/philosophy through his undergraduate days and graduate school in Rome. His competency with Thomism led him to defend its merit and purposes (Braun 1968). Before a crowd of Catholic philosophers, the Monsignor revealed his long-standing enthusiasm for neo-scholasticism. He argued: "The basic ideas of scholasticism are

living truths—firm enough to support the whole fabric of knowledge yet flexible enough to allow for every addition of ascertained fact" (Pace 1926, p. 16).

Thomism permeated Pace's work. According to Ryan (1961a), the priest-psychologist "never [lost] an opportunity to co-ordinate the principles of [neo-scholasticism] with the best scientific achievements of the day" (p. 148). Essentially, Pace possessed a philosophical paradigm deeply entrenched in Christianity while he worked as a psychologist—teaching, researching, and championing for the discipline.

### 3.2. Progressivism

From 1890 to 1920, progressivism was a nationwide reform movement in the United States. Societal ills resulting from rapid industrialization, urbanization, and immigration needed attention (Leonard 2011). Monopolies were crushing competitors and political corruption was prolific. Factory owners exploited children by demanding them to work long hours, receive little pay, and risk serious injuries. Hence, politicians (e.g., Theodore Roosevelt), academics (e.g., John Dewey), social activists (e.g., Susan B. Anthony, Jane Addams), clergy (e.g., Rauschenbusch), and journalists (e.g., the Muckrakers) endeavored to 'make America better.'

Several social, political, economic, and religious agendas emerged leading to the suffrage movement, anti-trust legislation (e.g., Sherman Antitrust Act), child labor protection (e.g., Keating-Owen Labor Act), new educational requirements, amendments to the Constitution (e.g., 16th–19th), and religious activism (e.g., Social Gospel) (Zainaldin and Inscoe 2008). In general, progressives advocated for big government to be the 'national' problem-solver that would enact and enforce solutions to America's problems. They "believed in the epistemic and moral authority of science . . . [meaning] intellectuals should guide social and economic progress" (Leonard 2011, p. 430).

Among Catholics, pastoral and academic leaders cheered for several of the aforementioned social reforms. Yet, they objected to and rejected the diminution of divine authority. Explaining the Catholic response, Woods (2004) noted, "In an age [leaning to] man-centered morality and emancipation from the dogmas of the past, Catholics [held the] only satisfactory answer to moral chaos was that provided by the Church" (p. 156).

Regarding Edward A. Pace, his teaching, research, and apologetic efforts attracted criticism. Critics labeled him a 'modern, liberal, and a progressive' and even called for his resignation at the Catholic University of America (Gillespie 2001). Messmer (1896), once a colleague, retorted to Pace that the professors at CUA were not "looked upon with favor by many Catholics" (p. 1).

Yet, the priest-psychologist was far from being a progressive extremist. He held tightly to and defended Catholic doctrine. His attachment to the scientific method of experimental psychology drew much ire. According to Elias (2005), "What made [Pace] progressive was the fostering of critical thinking," advocacy of methods that encouraged his students to question knowledge, and an opposition to "rote catechetical training" (p. 14).

### 3.3. Empirical Psychology

In the same year Pope Leo XIII promulgated *Aeterni Patris* in Rome, Wundt established the first laboratory dedicated to the investigation of psychological phenomena in Leipzig, Germany. Wundt applied the scientific method creating a measurement orientation in the discipline (Pickren 2000). The development generated considerable excitement. Bingham (1896) shared, "Every niche and crevice of the mental world can be searched and compelled to yield its treasures . . . [giving] better rules for all who deal with the mind" (pp. 350, 352).

The beginning was not easy for the new psychology. Pace (1894) remarked, "The undertaking was not without its difficulties. Space was wanting, appropriation was slow, and some of the on-lookers shook their heads in doubt" (p. 534). Yet, this psychological advancement was in step with many technological and industrial advances sweeping the United States and Europe. Revolutionary

developments in transportation (e.g., railroads), communication (e.g., telegraph), and medicine (e.g., brain and sensory functions) elevated scientists' social and political influence (Leonard 2011).

When Pace attended the University of Leipzig for the psychology doctorate, the discipline was only ten years old. Like other sciences, empirical psychology "followed the pattern of professionalizing" with the developments of journals, laboratories, and national organizations (Coon 2000, p. 85). In addition, the psychological revolution set in motion an intense rivalry between empirical psychology and rational or philosophical psychology.

According to Maher (1915), rational psychology examined "the source of the phenomena of consciousness [via] the light of reason" (p. 459). The 'soul' was the central subject matter (Ross 1992). Divinity and philosophy professors were the key educators of these courses in the American academy (Coon 2000). Some universities designated philosophical psychology as a 'capstone' course in their curriculum with the intentions of preparing students for future leadership positions (Pickren 2000).

At the heart of the rivalry was the notion that scientific psychology would supplant its elder relative. Pace responded to this concern. He wrote:

> The new psychology, while it advanced steadily to autonomy, by no means discarded as a whole, the whole acquisitions of the past. On the contrary, without the development that preceded, the modern state would have been impossible. The new psychology, though a transformation, is really a complement of the old. (Pace 1894, pp. 523–24)

The priest-psychologist appreciated and respected the elder version. Empirical psychologists were indebted to rational psychologists; exterminating its older relative was counterproductive. Often, in lecture, he iterated, "Thoughtful men are pretty well agreed on two things—first, that science needs philosophy, and second, that philosophy needs science" (Pace 1895b, p. 550).

Many of his psychological peers differed. The Monsignor recognized, "That to a large number of intelligent persons, metaphysics is distasteful, while within a narrower circle, philosophy is regarded as a drag upon science, a hindrance to its progress" (Pace 1895c, p. 166). However, supplanting notions seemed nonsensical. Psychologically-trained people cannot extricate themselves entirely from philosophy. Researchers benefitted from the aid of philosophy in solving problems. They also utilized philosophical terms (e.g., cause, effect) and required philosophical reasoning for their interpretations of research findings (Braun 1968).

## 4. Emulative Habits of Pace

Albert Bandura delineated characteristics of effective models. People imitate others whom they perceive as competent, prestigious, and relevant (Ryckman 2009). Even in a fortuitous event (e.g., reading an article about a first-generation psychologist), a model "can leave lasting effects [and] thrust people into new life trajectories" (Bandura 1999, p. 159). Some of the behavior of Edward A. Pace embody the three Bandurian traits and merit emulation. Four specific habits of the Monsignor warrant examination.

### 4.1. Appetence for Wisdom

Dr. Pace was a lifelong learner. In an article memorializing him, Yearly (1938) described the priest-psychologist as a "lover of truth and a 'truster' of the Truth-giver. He welcomed knowledge from whatever source it came" (p. 9). In the first habit, an appetence or hunger for wisdom in multiple disciplines, we explore Pace's educational journey in three disciplines and some of his experiences as professor and researcher.

### 4.1.1. Educational Training

Pace's theological training lasted ten years beginning in 1876. The journey started in his undergraduate days at St. Charles College in Maryland and continued in seminary at the Pontifical North American College (PNAC) in Rome, Italy. Completion of the bachelor's degree in 1880 at St.

Charles allowed him to accept the appointment to PNAC to prepare for priestly duties. Within the pontifical university system, Pace earned another bachelor's; this one was in Sacred Theology (STB) in 1883. He finished the doctorate in Sacred Theology in 1886 at the age of 25 (Misiak and Staudt 1954).

Often, the STB entails course work in systematic, sacramental, and moral theology along with scriptural classes in New and Old Testament. Students would also need to demonstrate competency in New Testament Greek and Latin (dhs.edu 2019a). A seminarian completes the doctorate in Sacred Theology with additional classes, most likely ones specializing in Thomistic theology, along with the creation, defense, and publication of a dissertation (dhs.edu 2019b).

Philosophical training transpired during the same decade of theological education. Pace (1895c) shared, "The wishes of the Sovereign Pontiff [was] a two year course in philosophy" (p. 167). Current pontifical STB programs expect about twelve to fifteen classes in philosophy. Coursework includes "logic, philosophy of nature, metaphysics, philosophy of knowledge, philosophical anthropology, philosophical ethics, ancient philosophy, introduction to Thomas Aquinas, medieval philosophy, modern philosophy, and recent philosophy" (dhs.edu 2019a).

Training in philosophy continued in Pace's doctoral program in Sacred Theology. Pope Leo XIII noticed his keen philosophical mind. Near the end of his time at PNAC, the Monsignor and a peer had been selected to participate in a debate at the Vatican presided by the pope. O'Connell (1931) recalled: "Both had distinguished themselves in Metaphysics. They knew extremely well the doctrines of Saint Thomas (*Contra Gentiles*) . . . They also knew Latin, the language the disputa [debate] was carried on" (p. 321). After the doctoral program, Pace also included philosophy courses when he registered for science classes at the University of Louvain and Sorbonne (Braun 1968).

Moreover, four years into his professor role at CUA (i.e., 1895), the Monsignor began teaching philosophy classes, an assignment lasting until 1935. Hence, Pace ardently defended the importance of philosophical training. He believed the development of a 'philosophical habit' empowered individuals to tackle problems and information. As a result, people would be more adept at conceptualization, interpretation, and discernment of issues and challenges. Since this "mental habit" was a vital component in intellectual maturity, Pace even advocated for philosophical training at the "lower grades" (Pace 1927, p. 23). Last, the priest-psychologist became a founding member of the American Philosophical Association and the Washington Society of Philosophy and Psychology along with being one of the founders and editors for the journal, *The New Scholasticism* (Braun 1968).

As for psychological training, the idea for additional scientific education beyond Louvain and Sorbonne had a spontaneous beginning. Browsing in a Paris bookstore, Pace fortuitously found a used copy of Wundt's book, *Principles of Physiological Psychology* (Sexton 1980). His interest for the new science was immediate, prompting him to request an extension to study the modern version of psychology at the University of Leipzig. With Bishop/President Keane's permission, Pace traveled to Leipzig. He quickly became acquainted with Professor Wundt, received advice on class registration, and then, secured housing on Rudolf Street about a half mile from the Psychology Institute (Pace 1889, 1921).

In a letter dated 26 November 1889, to Bishop/President Keane, the Monsignor reflected on his experience with the psychological education. Pace (1889) wrote:

> My work here in Leipzig is continually growing. 'Experimental psychology' is a sort of short way of summing up some dozen branches of science. It is bound to take the front rank in its own line, before many years, and judging from the number of Americans who are working at it here, it will excite a good deal of attention in our country. My scheme now is to take the Doctorate of Philosophy here. This would bring me into relation with the leaders on the scientific side of my specialty . . . I am convinced that to reconcile our philosophy with science, it is necessary to win a place in the respect of scientists.

The decision to study under Wundt was a courageous move. Many Catholic intellectuals in the United States were skeptical of this modern version; "experimental psychology was far from being accepted" (Misiak and Staudt 1954, p. 69). In his 26 November 1889, letter to Keane, the Monsignor

also referred to the controversy as it related to his submitted article on hypnosis. One month later (December 28), he wrote, "I am glad you [Keane] didn't publish my article. It might call out comments of a hostile nature, or lead to controversy which I have not just the time to sustain" (Pace 1889).

Pace completed the requirements for the psychology doctorate in two years. He enrolled in several psychology and physiological courses. One particular class, *Experimental Psychology* taught by Wundt in the winter semester of 1891, included E. B. Titchener, a British student who became a professor of scientific psychology at Cornell University and a major advocate for Structuralism in the United States (Belegbogen 1891). Ironically, the Monsignor wrote a philosophical dissertation as the capstone to his scientific psychology education; he titled it, *The Principle of Relativity in Herbert Spencer's Doctrine of Psychological Evolution*. Last, while in Leipzig, Pace frequently traveled back to Louvain, Belgium, to attend Mercier's lectures, possibly to find common religious ground with a respected mentor of science (Braun 1968).

### 4.1.2. Professional Roles

In his role as professor, Pace contended psychological knowledge offered previously unknown insight about the human condition. He opined:

In general, of course, it has always been known that there was some sort of connection between the psychical and the physical; what modern psychology accomplishes is the more detailed and exact investigation of that connection. Similarly, with the phenomena of association, memory, attention, inhibition, and fatigue [sic]; their importance has long been recognized, but their thorough analysis is the outcome of experimental work. (Pace 1902b, p. 22147)

Furthermore, his optimism regarding what 'modern' psychology had to offer his students and even the world was palpable. The Monsignor disclosed:

The new terminology [in psychology] suggests something tangible where all has hitherto been shadowy and vague. It hints at the unveiling of mysteries which have baffled the philosophers of the past. It awakens the hope that science may eventually devise formulas and laws for the world of mind as it has devised them for the world of matter. (Pace 1894, p. 522)

Originally, Bishop/President Keane hired Dr. Pace to teach philosophy. Instead, with the Ph.D. in experimental psychology, the Monsignor dedicated his time to the development of psychology classes. In 1891, the first psychology course offered to students (i.e., graduate level, Catholic clergy) at CUA was titled, *Experimental Psychology, Rational Psychology, General Psychology*. The content examined physiological (e.g., nervous system structures and functions), anthropological (e.g., human constitution—soul, body), and biological topics (e.g., animals, plants) (CUA 1889).

According to Johnson and Jones (2000), Pace was an exemplar first-generation psychology professor. He chose to supplement lecture content with theological material (e.g., the soul). In a memorial for the Monsignor, Smith (1938) testified about the priest-psychologist as an educator. He shared Dr. Pace's success in the classroom was due to "his innate courtesy, his simplicity of heart that established sympathetic contact so quickly, and his extraordinary ability to encourage" (p. 2). Other psychology classes taught by Pace included *Theories of Mental Evolution and Abnormal States* in 1893 and *Individual Psychology* in 1894 (CUA 1889).

As a researcher, Dr. Pace organized a psychological laboratory occupying eight rooms in McMahon Hall on the CUA campus (CUA Archives 2016). This space included a lecture room, library, reading room, experimental rooms, and the professor's study. In addition, several instruments, similar to the ones that Pace utilized at the University of Leipzig, were available for experimental activities (Institute 1909). These included tuning forks, chronoscopes (time-reactions), kymographs (recorder of fluid pressure), plethysmographs (recorder of volume changes), and phonometers (sound intensity) (CUA 1889).

The Monsignor expected his students to practice the research process with him. Together, they followed the inductive steps of the scientific method. "Laboratory work teaches one the importance of details and how details fit into larger more complex problems" (Braun 1968, p. 83). He believed research methods prompt and train people to become critical thinkers.

Some of the research conducted in the psychology lab in McMahon Hall went to publication (Misiak and Staudt 1954). Pace published articles on visceral disease and pain (Pace 1897), binocular rivalry (Pace 1901), and fluctuations of attention and after-images (Pace 1902a). The Monsignor also pursued membership in the American Psychological Association. Although he was not a charter member, Pace was "among the first five psychologists elected by the charter members" (Ross 1992, p. 140). Furthermore, his research experience and reputation spread beyond CUA. In the months leading up to the 1904 Universal Exposition in St. Louis, Hugo Muensterberg invited Dr. Pace to provide leadership for the Experimental Psychology section (Pace 1904).

Last, it is important to note that Pace's research endeavors joined other empirical psychologists who provided the world in the 19th and 20th centuries with valuable data about the mind—the cognitive, emotive, and volitional functions. In 1894, a journal editor decided to publish articles from an advocate and critic of the new psychology. Pace, as researcher-apologist, seized the opportunity to mention the abundance of 'mind' research. During his examination of the literature, he found 900 articles published in 1889; 1325 in 1890; 1171 in 1891, and 1185 in 1892. Furthermore, the Monsignor highlighted recent advances achieved from the research. Progress had been made in the analysis of sense perception (e.g., taste, smell, hunger, thirst), the study of attention (e.g., mental fixation), the succession of mental states (e.g., duration of mental processes), the time-sense (e.g., shorter or longer intervals), and feelings and emotions (e.g., gladness, sorrow). Evidently, psychological researchers were "unveiling mysteries" and "awakening hope" (Pace 1894, p. 75).

### 4.2. Keen Awareness of Rival Worldviews

Dr. Pace was an informed scholar with highly developed discernment skills. Discussing an effect of a 'philosophical mind,' Braun (1968) commented, [People (e.g., the Monsignor, his students) are] "more capable of distinguishing theories and systems that ultimately dehumanize [them] from those in accord with [their] dignity as intellectually endowed person[s]" (p. 18). In the second habit, a keen awareness of rival worldviews, we examine some of the contemporary opposing worldviews active in the priest-psychologist's era and his awareness of and response to the paradigms.

### 4.2.1. Contemporary Rivals

Several contemporary professionals of Pace's era embraced an ideology rivaling Christian theistic values and beliefs. Specifically, their paradigms repudiated metaphysical realities associated with the human condition and reality. Wundt (1892) wrote, "The idea of the spirit being a sensible being separable from the body . . . was scientifically relegated to primitive races" (p. 104). James (1896) concluded in his *Principles of Psychology*, "Our psychology will remain positivistic and non-meta-physical . . . The substantial Soul explains nothing and guarantees nothing. I feel entirely free to discard the word from the rest of this book" (pp. 182, 350). Williams (1899) shared in the popular *Harper's Monthly*, "The partially liberated spirit of the new psychology had by no means freed itself altogether, at the close of the first quarter of our century, from the metaphysical cobwebs of its long incarceration" (p. 515). Last, Freud (Freud 1964) maintained, "Religion is a system of wishful illusions together with a disavowal of reality, such as we find nowhere else but in a state of blissful hallucinatory confusion" (p. 71).

### 4.2.2. Awareness and Responses

Discernment of worldview differences was an important topic of discussion for Dr. Pace. His lectures and publications reveal an astute perception of rival paradigms, ones that challenged and bothered the Catholic critics, his students, and himself. In the Monsignor's first psychology class,

there was a set of lectures on the nature of the soul. The professor-apologist began the section by differentiating well-known, philosophical systems espousing ideology on the soul. On one view, actualism, he shared:

> The Actualists [e.g., William James who promoted stream of consciousness] would say, 'We don't do away with the soul altogether. We simply give it new meaning. We can't allow your meaning of the soul, given to it by Aristotle, your great St. Thomas and the Scholastics.' The whole school of Actualists dread and avoid the word 'substance' as though it were the cholera, with a scrupulousness, which would make them all saints in the moral order . . . Why doesn't the poor man say soul and be done with it. (Pace n.d.)

In an article, the Monsignor expounded on two other worldviews with soulish ideas. He explained, "The materialist strives to show all mental phenomena are essentially organic; the spiritualist, aiming at a different conclusion, insists upon the distinction between lower and higher operations" (Pace 1895a, p. 145).

Pace, responded to religious criticism about modern psychology predisposing people to anti-God ideology. He countered:

> The question of [experimental psychology's] materialism, idealism, or spiritualismis quite independent of individual opinion regarding the nature of mind. The blame attaches not to empirical research, but to the metaphysics of the individual. Experimental psychology occupies a neutral position. Per se, it is neither monistic nor dualistic, neither materialistic not spiritualistic. It leans neither to one side nor to the other . . . It does not undertake to say what the soul is or what it is not. (Pace 1895a, pp. 142, 145)

In addition, the Monsignor opined, "Should some folk become materialists after studying experimental psychology, the trouble is with the students and not with the study, with their lack of logic and not with the principles of research" (Pace 1895a, p. 155).

Dr. Pace was far from naivety regarding paradigms that were contradictory of Christian doctrine. He agreed in part with the religious critics; several first-generation psychologists and pundits possessed a naturalistic worldview. Unlike the Catholic dissenters, the Monsignor reminded his audience that knowers—scientists, theologians, etc.—have biases; "their metaphysics" mold their knowledge. This comment has a hint of 'post-modernism.' People cannot separate themselves from their limitations and opinions. However, the researcher-apologist over-reached. He remarked that the 'field' of empirical psychology took a neutral position on ideology; it was "neither materialistic nor spiritualistic" (Misiak and Staudt 1954, p. 77). In addition, Pace regarded un-interpreted psychological data as neutral (Braun 1968).

Any field of any discipline would be comprised of knowers who were not 'neutral' or detached from their knowledge (Atkinson 1990). Scientists would have selected research questions and ascertained psychological data under the influences of their own theoretical assumptions. Advocates of naturalism would have assumed reality resides in nature and denied the dualism of matter and spirit, while theists most likely embraced supernaturalism and the existence of metaphysical realities (Gillespie 2001; Knight 2017). Possibly, 'modernistic' ideology, prevalent in his day, leaked into Pace's statements.

### 4.3. Intentional Ripostes to Catholic Critics

Dr. Pace was a bold defender. He refused to ignore the criticisms about modern psychology (Braun 1968). In the third habit, intentional ripostes or thoughtful responses to Catholic critics of scientific psychology, we discuss common disparagements lodged at the newest version of psychology and selected apologetic responses by Pace.

4.3.1. Common Disparagements

When the priest-psychologist returned from Europe in 1891, he was quickly embroiled in a firestorm of derision. The scope of the controversy is difficult to ascertain. Yet, the circulation of religious criticism toward the new psychology motivated several of the Monsignor's contemporaries (e.g., G. Stanley Hall) to function as apologists using popular media outlets (e.g., Harper's Monthly, The Forum). The mission was to persuade the American public that the measurement orientation of empirical psychology was legitimate and non-threatening (Pickren 2000). For Pace, his apologetic efforts with a similar mission targeted interested parties in Catholicism through church related media channels (Elias 2005).

Several professors teaching philosophical or rational psychology in Catholic academies resisted the emergence and recognition of the scientific version of psychology. Ironically, the pope expected these educators to be competent in science. Pace (1895a) remarked, "Pope Leo XIII, besides laboring in many ways by word and deed to quicken the spirit of research among Catholics, says that they should take up the study of science" (p. 160). However, skeptical academics perceived the new psychology as a threat, a menace whose romance with modernity might exterminate its older relative (Gleason 1995; Misiak and Staudt 1954). They also seriously doubted any compatibility between scientific psychology, rational psychology, and Catholicism.

Three notable disparagements were circulating in the Catholic academies during the late 1800s and early 1900s. Interestingly, there are similar counterparts in the 21st century among Protestants. First was an anthropological complaint. Catholic faultfinders described scientific psychology as a *soul-less approach*. Hughes (1894), a professor at St. Louis University, opined, "A man must be willing to admit that his soul is not more spiritual than his eye ... [I] prefer a psychology that includes mind and soul" (pp. 801, 812). Grunder (1912), another professor at St. Louis University, maintained, "Psychology has come to discard the soul ... [It] is essentially a psychology without a soul" (p. viii).

These critics argued the non-recognition of a spiritual soul (i.e., a metaphysical reality) and the over-emphasis on embodied psychological functioning were dehumanizing steps (Misiak and Staudt 1954). Interestingly, there are 21st century Protestant critics espousing similar anthropological concerns. Lambert (2016) declared, "The inability of secular therapy to acknowledge that mankind is made with an immaterial soul cripples its ability to offer meaningful counseling care" (p. 10).

Second was a philosophical objection; empirical psychology was *grounded in materialism*. Hughes (1894) expounded, "The materialistic theory seeks to explain the mind through the functions of matter, instead of all life-processes as having their origin in the self-activity of souls" (p. 803). Grunder (1912) concluded, "We live in an era of materialism. Psychology has taken a materialistic turn" (pp. vii–viii). In the 21st century, some Protestant critics also have philosophical complaints. Powlison (2010) argued, "Secular psychologists can't help the godlessness of their view of the psyche. Theories systemize and rationalize the unbelief of those who create the theories" (p. 278).

Third was a theological protest; the new psychology was *impious* and *dangerous*. Hughes (1894), pronounced "And if they [empirical psychologists] really mean to subject psychological activity to laboratory investigations, we do not scruple to call the whole enterprise a theological impiety" (p. 800). In other words, empirical psychology was an irreverent approach, disrespecting God. Moreover, anyone associated with the modern version was suspect and untrustworthy. In 1896, Bishop Messmer agreed to have Pace speak at the Columbia Catholic Summer School in Wisconsin. Yet, the professor's content was noticeably limited. Messmer (1896) wrote, "We shall consent to your lecturing only on the clear understanding that you will not bring up any matter in connection with your subject" (p. 2). In the 21st century, some theological complaints exist as well. Lambert (2016) wrote, "Christians who insist on employing secular language to describe counseling problems, rather than biblical language have not submitted to the authority of God's Word. [They] are guilty of serious error" (p. 12).

4.3.2. Pace's Intentional Ripostes

In his four decades of service at the CUA, Edward A. Pace directed a large portion of his time and energy responding to the aforementioned disparagements and other criticisms. His role as apologist was self-designated and an unpaid position. Like other first generation psychologists, he designed a rhetoric to persuade a predominately-Catholic audience through various mediums—popular and scholarly literature, lectures, and speeches (Pickren 2000). Consistently, the Monsignor argued empirical psychology and its methodology were legitimate, congruous, and innocuous. Three ripostes illustrate his apologetic efforts.

Regarding the anthropological complaint, the professor-apologist remarked, "'Psychology without a soul' is a phrase often misused to brand and presumably to crush, the audacious offspring before it is old enough to speak its defense" (Pace 1894, p. 523). Pace apparently considered the statement a dramatic message; its purpose was to spread antipathy concerning the new discipline. In response to the complaint, he remarked that experimental research in psychology does not include or exclude the soul; its foci are mind stuff. In the Monsignor's words:

> Experimental psychology furnishes accurate data regarding mental processes. It has nothing to do with ultimate causes directly ... [And,] the experimental method is necessary for every psychologist who believes that the workings of mind deserve thorough investigation".
> (Pace 1895a, p. 141)

Moreover, according to scholasticism, a mental operation such as sensation, "is a function, not of the soul alone, nor of the body alone, but of the body-soul composite" (Pace 1895a, p. 152).

Concerning the philosophical objection, the priest-psychologist retorted, "Exaggerated as these notions [other criticisms] may be, they are less unjust than the opinion which regards the new science as a finer form of materialism" (Pace 1894, p. 523). What was unjust is the illogical reasoning of the religious critics. Indeed, the methods of experimental psychology were limited to the material side of mental functioning. However, this does not mean researchers embraced the philosophy of materialism. The Monsignor inquired, "Can we, without becoming materialists, vary the conditions of mental activity, note the effect of each variation, and hence determine what causes are at work?" (Pace 1895a, p. 149). In other words, the argument—use of methodology automatically leads to the embrace of a particular philosophy—was specious in nature (Braun 1968).

Pertaining to the theological protest, Catholic disparagers stereotyped scientific psychology, its adherents, and its methodology as a field of disrespect and dishonor to God. Impiety was a serious accusation. If the stereotypes were valid, was there evidence for the claims of impiety—particularly for the Monsignor? Supporters of Pace would have disagreed; their memorials were a stark contrast to this criticism. Sheehy (1938) attested to the priest-psychologist's loyalty to the church and his faith and confidence in God. Smith (1938) remarked how Pace was obedient to the church and respectful to his authorities. Brennan (1938) underscored his gentle, kind, modest nature—qualities that brought honor to God. Regarding the reputation of CUA, Dr. Pace affirmed his colleagues' fidelity to God and Catholic doctrine. He disclosed:

> I am convinced that this University, as a whole, and in its individual members, is, and always has been, loyal to the teaching of the Catholic Church. I know that accusations have been made against it, or against some of its professors. It has been charged with 'liberalism' and even with materialism; but I have never seen any evidence of such tendencies. There has certainly been no 'modernism' in the sense condemned by Pope Pius X. (Pace n.d.)

*4.4. Unrelenting Steadfastness to the Christian Faith*

Dr. Pace was a faithful follower of Christ. Although his tenure at CUA entailed several daunting challenges and setbacks, the Monsignor remained firmly anchored to God. In the final habit, unrelenting steadfastness to the Christian faith, we review samples of the priest-psychologist's doctrine and his unique position within the neo-scholastic community.

4.4.1. Doctrinal Sample

Catholic critics not only disparaged scientific psychology; their faultfinding became personal and extended to adherents' Christian faith and fidelity to the Church. Hughes (1894) remarked, "A man would have to forswear his belief in a truth of Christian faith … if he wishes to have anything to do with the 'new psychology'" (p. 801). Unfortunately, there was some historical warrant for Hughes's over-generalization. A few, well-known, psychologically-trained Catholics (e.g., Franz Brentano) became apostate (Misiak and Staudt 1954).

As mentioned previously, two influential authority figures in the Catholic Church directly challenged the Monsignor's loyalty. Satolli, an apostolic delegate, recommended to Pope Leo XIII that Pace should be dismissed from CUA (McAvoy 1957). Messmer (1896), a bishop, limited Pace's speaking content for an engagement. Both men insinuated in their decisions that the priest-psychologist was unfit and doctrinally unsound.

A sample of the priest-psychologist's doctrine disputes these conclusions. Two specific doctrines attest to his faithfulness to Catholicism and God's Word; each illustrate how Christianity flowed into his psychological world. First is biblical anthropology. In the late 19th century, a central focus of scientific psychology was mental processes. The professor supplemented scientific psychological information with anthropological material in the second semester of his first psychological course at CUA (Johnson and Jones 2000). His lecture notes included views on human constitution—in particular, a description of the soul, how it relates to the body, the soul's origin, and immortality. Pace (n.d.) expounded:

> [The soul] is in us, the source of all our activities as organic beings. It is the source of all our conscious processes, in particular those of intelligence and will. The soul must cooperate with the organism. But, this soul does not depend for its existence upon, or borrow its existence from the body. [Moreover,] the soul is a spiritual substance and cannot originate from any material substance. It can't [derive from] the parental act or evolve out of the organism. The Creator [God] produces each human soul and [He] infuses [a soul] into the embryo. [Last], the soul does not perish with the body, but lives on after the body has ceased to live. [More specifically], at death the bodily elements fall away from their union with the soul. The soul continues on its own existence, with the larger part of the individuality unimpaired. (pp. 348, 392, 407, 410, 414, 416)

A second doctrine is theology proper. Since religious critics labeled empirical psychology as impious and scientific psychologists as dangerous, Dr. Pace's views on God the Father are noteworthy. The Monsignor spoke at the rally for the Feast of Thomas Aquinas in 1896. His message before the student body at CUA revealed several doctrinal views. He described God as a divine being who is immanent and transcendent and who willingly discloses His truth to humans via Scripture and creation. Pace (1896) shared:

> And thus, from the things [in nature] that are made, we rise to a knowledge of their invisible Maker. But, because God, their Maker, is a personal God, infinite mind and infinite will, in one; and because in the depths of unlimited being and unbounded goodness there are truths which no finite mind, of itself, can perceive; it is possible, it is fitting, that God should make Himself known … God manifesting Himself in the universe and God revealing Himself in His incomprehensible Word. (p. 192)

Smith (1938), in his memorial of Pace, mentioned a conversation that the two had shortly after the turn of the century. The Monsignor disclosed an ambition; he wanted to "die a good, old, simple priest;" Smith attested that Dr. Pace received his wish. "Old was this Apostle of Truth—seventy-seven years. Good was he in every way and true. A priest he was until the end and forever" (p. 2). Essentially, the priest-psychologist engaged in the new psychology and remained loyal to the faith defying Hughes's overgeneralization.

4.4.2. Neo-Scholastic Community

The Monsignor openly and proudly aligned himself with neo-scholasticism. The attachment started in his undergraduate days and lasted until his passing. Sheehy (1938) mentioned how he enjoyed hearing "the easy flow of Scholastic philosophy which was a part of his life" (p. 3). Yet, there are questions as to which stream within Thomism did the priest-psychologist align (Doyle 2007). Pace did indicate an awareness of an interpretative struggle. He commented:

> At a time when Thomistic philosophy is proposed to the world as a remedy for the evils wrought by so many systems, the first care of its advocates should be to speak the genuine language of the school whenever they espoused its doctrines. (Pace 1896, p. 140)

According to Kerr (2002), Thomism had always been in disputation. There were several opinions concerning Aquinas's work. He noted:

> In the versions of Thomism current from the 1850s to the 1960s, Thomas's work was regarded as a high point of medieval Christianity, either a unique balance of faith and reason, a harmonizing of revealed theology and natural theology, an incomparable synthesis, or (by adversaries) as a singularly vicious corruption of Christian doctrine by Hellenistic paganism. (p. 14)

Moreover, other Catholic theologians/philosophers identified specific interpretative problems. Gilson (Gilson 1988) underscored rationalism and deism. He declared, "The Thomist theology established in Catholic seminaries and universities was seldom other than 'rationalism': pandering to the 'deism' that most Thomists—'deep down'—prefer to teach" (pp. 23–24). Though unintended, neo-scholastics' counter-modern stance ironically reproduced the very outcomes they opposed.

In both distinctions by Gilson, there was a centralization of human reason and diminution of divine wisdom. Although God is a Creator in a deistic worldview, this divine being is transpersonal and dispassionate. Pace's views on God are far from Gilson's summary of what many neo-scholastics embraced.

As previously mentioned, the Monsignor characterized God the Father as an immanent being—One who sustains a relationship with the world. In his words, "God, their Maker, is a personal God ... Nor was God, an isolated, unknowable somewhat, entirety apart from the world (Pace 1896, p. 192). Simultaneously, the Lord stills functions as a mysterious, transcendent being who is omnipotent and infinite. The priest-psychologist spoke:

> A divine energy is put forth in every production of nature's causation ... each physical process is a manifestation of God's power ... Each effect is more truly the outcome of God's omnipotence than of the physical agency from which it immediately proceeds. God is of infinite mind and infinite will. (Pace 1896, p. 192)

God is also, "the efficient cause of man" (Pace 1895a, p. 137). In addition, this omnipotent being seeks to make Himself known to the world population. In Pace's words, "It is fitting, that God should make Himself known by a more immediate revelation ... God manifesting Himself in the universe and God revealing Himself in His incomprehensible Word" (Pace 1896, p. 192). Apparently, in the Monsignor's theology, the created order including humankind was not operating autonomously from God's activity.

Furthermore, Dr. Pace showered accolades upon human wisdom. He shared:

> The science of our day, my friends, fills us with admiration of Nature's beauty, because it shows us with the lens of observation and the crucible of experiment, how much is concealed beneath the humblest form and the simplest function. (Pace 1896, p. 193)

However, autonomous human reason was not centralized. The Monsignor knew that "many modern interpreters of Scriptures [were] of the rationalistic school" (Pace 1895a, p. 142). Instead, Pace discussed:

The unification of knowledge . . . a splendid unity. [More specifically]: The knowledge that comes by seeing and the Faith that comes of hearing, there is, and there can be, no suspicion of discord . . . We behold the most perfect blending of natural and supernatural truth . . . St. Thomas pointed the path to synthetic knowledge. (Pace 1895a, pp. 192–93)

The priest-psychologist also clearly declared limits in human reason. He stated:

What shall we say of the effort to harmonize both science and philosophy with the teachings of Faith? Reason is brought face to face with that which immeasurably transcends it. (Pace 1895a, p. 191)

Such rhetoric bends neo-scholastic rationalism and deism perhaps to the breaking point. Some historical reviewers still might apply 'guilt by association.' Close examination of Dr. Pace's words suggests otherwise. If Gilson's generalization was fair and appropriate, then possibly, the priest-psychologist was more of an 'outlier' in the neo-scholastic community of his era. Mostly likely, more than the Monsignor was aware.

## 5. Applications for Psychologically-Trained Protestants

The aforementioned habits of Dr. Pace have contemporary relevance for psychologically-trained Protestants. Over forty years, the Monsignor attempted to competently teach graduate students, research psychological phenomena, and defend a person of faith's place in scientific psychology. In this last section, we propose several suggestions for putting aspects of his lifestyle to use in a practical manner.

### 5.1. Pace's Appetence for Wisdom—Honor the Dominion Command

In Genesis 1:28, Yahweh declared, "Be fruitful and multiply and fill the earth and subdue it, and have dominion over [it] . . . " (English Standard Version; ESV). God ordained humans to be responsible stewards of creation (Atkinson 1990). Studying psychological phenomena is one of many means to honor the dominion command. Philips (1980) noted, "Every scientific advance, every new scrap of knowledge about the universe is an outworking of that dominion" (p. 46). Likewise, Robertson (1980) maintained that having dominion or subduing the earth "involves the bringing out of all the potential within the creation which might offer glory to the Creator;" it is a "creational responsibility" (p. 80).

To the best of our knowledge, Dr. Pace did not discuss or mention the above interpretation of Genesis 1:28 in his presentations or publications. Yet, his practices reveal an observance of the dominion command. Three notable examples illustrate.

First, as previously mentioned, the Monsignor respected and sought information from a variety of sources. He regretted not taking the time earlier in his life to learn scientific information (i.e., psychology, physiology). In a letter to Bishop/President Keane dated 6 March 1890, the priest-psychologist disclosed, "It is rather provoking to realize that I have grown old [he's 29 at this time] without even a suspicion of the riches of science and to see boys of twenty better equipped than I for the struggle of our day" (Pace 1890, n.p.). Furthermore, Pace also remarked, "The more deeply we penetrate the secrets of nature, the more keenly we analyze the laws, the workings and the products of human intelligence" (Pace 1896, p. 193).

Pace, apparently, gave himself freedom to pursue wisdom from sources outside the biblical and theological realms. Several of his publications included discussions on how psychological knowledge benefitted the Catholic Church, an outworking of the dominion command and an effort that intended to honor God (Braun 1968). Even in a zeitgeist where respected, religious voices spoke disapprovingly about scientific psychology, the Monsignor was undeterred. Likewise, 21st century psychologically-trained Protestants can robe themselves in the same freedom. They are unrestrained while embracing 'creational responsibility' and drawing out potential wisdom in God's created order as discovered in psychology (Robertson 1980). In real life settings, some psychologically-trained individuals have had no reason to apply the Genesis imperative in this fashion; they most likely do

not perceive themselves as restrained in any manner. Others understand what it is like to encounter a challenge or criticism by a friend or an unfamiliar individual; a biblical rationale for involvement in the field can have merit. Honoring the dominion command might become one of several helpful 'talking points' in discussions with detractors.

Second, Pace refused to discard psychologists' findings due to disagreements with their worldviews, and he promoted humble, gracious responses to their opinions. He stated:

> Let it be granted at once that some psychologists have gone astray in their philosophical deductions. We may criticize and cast aside their conclusions, but the error which these contain does not destroy the facts from which they are drawn. (Pace 1895a, p. 144)

Furthermore, the Monsignor noted:

> Feeling how limited is our own span of truth, we will readily make allowance for those whose opinions we cannot logically endorse. We will gain their respect and mayhap their love. (Pace 1896, p. 193)

The priest-psychologist had an obvious desire to respect critics—redeemed and unredeemed professionals. Even though their interpretations of psychological professionals opposed his worldview, Pace maintained research findings could generate a part of human wisdom (Pace 1896).

The Monsignor was seemingly comfortable working with knowledge developed from and framed in a rival worldview. He was not naïve about or careless with the influence of rival philosophies (e.g., agnosticism, materialism, naturalism, evolution). It is possible his comfortability developed from a secure attachment to the Christian faith and a dependence on the Holy Spirit for discernment. Likewise, 21st century psychologically-trained Protestants can function with the same comfort. In real world applications, some are untroubled by rival worldviews; they have not had a reason to doubt their loyalty to the living Christ. Others have experienced challenges regarding their fidelity. These inappropriate assumptions about disloyalty are reckless ad hominem tactics. The Apostle Paul reminded followers of Christ to be 'steadfast and immovable'—not easily disturbed by or prone to shifting due to false accusations (1 Corinthians 15:58) (Morris 1976).

Last, Pace advocated the combination of wisdom discovered from careful analyses of the created order (general revelation) and from illumined interpretations of inspired Scripture (special revelation). In a presentation to CUA students, the Monsignor addressed the importance of 'synthetic knowledge.' He shared:

> What shall we say of the effort to harmonize both science and philosophy with the teachings of faith … In his [Thomas Aquinas] mind, as expressed in his writings, we behold the most perfect blending of natural and supernatural truth … We find a model of synthetic comprehension". (Pace 1896, pp. 191–93)

Although the priest-psychologist utilized different terminology and never published examples of integrative outcomes, it appears he confidently promoted integrative possibilities (Misiak and Staudt 1954). He assumed, "The knowledge that comes by seeing and the faith that comes of hearing, there is, and there can be, no suspicion of discord" (Pace 1896, p. 192). Likewise, 21st century psychologically-trained Protestants can make that similar assumption and boldly pursue synthetic potentials between psychology and Christianity/theology where it is appropriate as another means to honor the dominion command.

### 5.2. Pace's Keen Awareness of Rival Worldviews—Recognize Risks and Resources

Psychologically-trained Protestants in the 21st century have immersed themselves in a scientific community whose epistemology, in general, is like a different language (Johnson 2007). Their native tongue embraces distinctively different assumptions about "what is real, what is good, who [humans] are, and what [humans] should do" (Willard 2009, p. 45). Being fluent or competent in psychology

while remaining fluent and faithful to theology or Christianity requires intentionality. Interestingly, some contemporary Protestant critics have confidently concluded balancing fluency in both is almost impossible. Powlison (2010) warned:

> Those who overweigh the significance of secular psychology 'learn' more than they bargained for. They tend to undergo a wrong-way conversion, become anesthetized to the God-centered realities actually playing out in the human psyche. They begin to reason godlessly about behavior, mood, relationships, motives, cognition, and so on. They promulgate faulty reasoning and practice through the body of Christ ... The Bible becomes an ancillary and supportive text, a source of proof-texts in the worst sense". (p. 284)

Risks exist whenever followers of Christ in the 21st century seek to balance vibrancy in their faith and competency in any discipline (e.g., psychology, theology). Any redeemed individual can become imbalanced or seduced to a contrary epistemology; he or she could have a 'wrong-way conversion.' The term, 'seduce,' refers to being persuaded toward disobedience or being un-loyal (Seduce 2017). The Apostle Paul warned in Colossians 2:8 that Christ-followers should not be taken "captive by philosophy and empty deceit" (ESV). The apostle wanted his readers to "look carefully [in order to not be] carried off as plunder by teaching devoid of intellectual, moral, or spiritual value" (Moo 2008, pp. 185–86). The image is an articulate speaker gathering an audience impressed by his or her message and sweeping them away for induction [into some kind of group] (Dunn 1996).

Yet, two powerful resources can assist psychologically-trained Protestants in the 21st century when they strive to discern rival worldviews and pursue balance—competency in psychology and faithfulness to Christianity. The first resource is the indwelling Holy Spirit. Christians in the psychological world can rely on the third person of the Trinity to bring about maturity—a shrewdness that can identify truth in psychological phenomena and thwart seduction. The author of Hebrews claimed 'maturity' results from practice; "powers of discernment" of redeemed people are "trained by constant practice to distinguish good from evil" (5:14; ESV). According to Bruce (1990), mature Christian spiritual faculties are honed via life experiences creating a "standard of righteousness" enabling them to discriminate "moral situations as they arise" (p. 136). This capacity of discrimination can serve believers as they study, research, or apply psychological phenomena generated by psychological professionals without a theistic worldview or value system (Lane 1991).

Some contemporary Protestant critics have asserted psychologically trained Christians tend to "discount the pervasiveness of the Fall" (Genesis 3) by not viewing psychological phenomena as part of the "tainted creation" (Hardin 2014, p. 324). Indeed, the critics are partially correct. Psychological knowledge has been infected with sin; it also possesses limitations (e.g., bias, finiteness, the unknown, misperceptions). However, all human knowledge (e.g., theology) is in the same predicament. The choice to pursue information from non-biblical/theological sources may not be a discounting of the doctrine of the Fall. Instead, it might be more indicative of modern-day religious individuals not limiting their resources of knowledge. Like Dr. Pace, they possibly have granted themselves biblical freedom because of the dominion command (Genesis 1:28) to pursue varied sources of wisdom.

Furthermore, authors' rival worldview beliefs embedded in psychological phenomena are discernible. Detection will not always be easy, but it is also not always impossible. For instance, Carl Rogers rejected innate human tendencies for harmful/sinful behaviors and promoted unconditional positive regard as a condition for client change (Ryckman 2009). The former contradicts the biblical message on sin nature; the latter partly mirrors a 'divine-love' prompting therapists to love their neighbor (i.e., client) as themselves as a criterion for mental health (Mark 12:31). In this example, psychologically-trained Protestants are not limited to binary options—embracing or rejecting all of Rogers' ideas. Such a perspective mimics the rigidity in dichotomous thinking, which sometimes torture religious pundits. Instead, they can offer unconditional positive regard. This may entail genuinely valuing clients and respecting their self-determination (i.e., agency) while disavowing Rogers' anthropological assumption of no innate tendencies for harmful behaviors and remain true to their theistic values. Hence, the discriminating skills and the worldview of 21st century Protestant

users of secular knowledge are paramount. They can remain faithful to their theistic values and beliefs while using or integrating psychological phenomena into their research or therapeutic interventions or any application.

The second resource is critical thinking or the 'philosophical mind' as espoused by the Monsignor. He strongly suggested in a letter dated 2 September 1890, to Bishop/President Keane, "Let the students know their opponents as well as their friends: teach them to read critically—they will learn to think for themselves and will not be trapped by sophistry in their after-studies" (Pace 1890, n.p.). Scriven and Paul (1987) marked critical thinking skills as evaluating, analyzing, synthesizing, and applying information. Such cognitive abilities empower individuals to generate knowledge. Willard (2009) reminded learners, "Knowledge involves truth or accuracy of representation based upon adequate evidence or insight. Knowledge [also] confers on its possessor an authority to act, supervise policy, and teach" (pp. 15, 17). Protestants in the 21st century can confidently engage psychological information by characterizing an author's worldview (analysis) and deciphering biased interpretations (evaluation) (Zawisza 2018). For example, naturalism clashes with its rival, theism. Naturalists espouse, "All knowledge is reducible to sense perception and there are not two radically different kinds of reality" (Willard 2002, p. 25). Theists believe in God who is both imminent and transcendent and who is an "existing self-revealing being. Cognitively justified belief in God aims at truth, at an accurate portrayal of reality" (Copan and Moser 2003, pp. 2–3).

### 5.3. Pace's Intentional Ripostes to Catholic Critics—Constructively Engage Protestant Critics

Critics of scientific psychology have not relented. Their protest has ensued for almost one hundred years. Catholic disparagement began in the 1880s while Protestant criticism became notable in the 1970s (Adams 1970). Generally speaking, topics of protest varied by group. Among Catholic faultfinders, disparagements tended to be global. In other words, much of the new version of psychology (e.g., its popularity, the scientific method, worldviews of adherents, anthropological views, etc.) was bothersome.

For Protestants, the scope was/is narrower. These skeptics tended/tend to target psychotherapy with two complaints, role reduction and seduction. Instead of the local minister counseling parishioners, the church out-sourcing this task to therapists was and still is problematic (Bulkley 1993). In addition, psychologized counselors were/are infiltrating Protestant churches with humanistic values that undermine biblical authority (Hunt and McMahon 1985; Powlison 2010). Lambert (2016) opined the 21st century church must have a reformation, a serious debate about counseling, similar to the charge Martin Luther led against indulgences. Among Lambert's ninety-five theses, secular counseling theories and interventions are identified as harmful, unhelpful, wrong, and insufficient. In addition, Christian therapists who use secular practices undermine the authority of Scripture and are guilty of malpractice.

Ascertaining how widespread criticism of scientific psychology in the 21st century is difficult to calculate. Yet, several spiritually mature, respected, and influential voices with access to media outlets are loudly dissenting. Powlison (2010) warned psychologically trained Protestants tend to experience a "wrong-way conversion" and become "anesthetized to God-centered realities" (p. 284). Bigney (2012) argued evangelicals have switched the "language of Scripture for the language of psychology" (p. 21). White (2014) purported redeemed counseling professionals are obliged to work in an unbiblical sense. Lambert (2016) stated, "Christians who have counseled without relying on the resources of Scripture should repent to those they have tried to help" (p. 13). In sum, Johnson (2010) observed, "The Christian [Protestant] counseling community has been experiencing something like a 'Forty Years' War,' in which various factions have criticized and denounced one another" (p. 311).

What are possible responses to comments of this nature? Influenced by Pace's lectures, articles, and speeches, we proffer two tactics for psychologically-trained Protestants in the 21st century. They can constructively engage critics by refusing to ignore disparagements and choosing to share a personal rationale for involvement in scientific psychology.

Ignoring critics' opinions is a tempting response. Given the nature of some views, a reply may seem futile or unworthy of the effort. Pace (1895a) referred to disparagements as unjust exaggerations and "chimerical monsters evoked to frighten people" (p. 162). Yet, dismissing criticism can be problematic for three reasons. First, erroneous perspectives about psychology among Christians only perpetuate if there are no counter-arguments. A non-response to criticism fails to defend the importance of a Christian presence within psychology in order to have a voice or sway in academic discovery. In other words, a cogent, respectful opposing view widens discussion. Pace (1895a) warned the Catholic Church:

> Either get hold of this instrument [scientific psychology] and use it for proper purposes, or leave it to materialists, and after they have heaped up facts, established laws, and forced their conclusions upon psychology, go about tardily to unravel, with clumsy fingers, this tangle of error. Either share in the development of the science, or prepare to wrestle with it when it has grown strong in hostile service. (p. 160)

Second, ignoring can overlook a prompt by the Holy Spirit to self-assess one's heart for any possible unbiblical assumptions (e.g., minimizing the sin nature). Third, dismissing criticism prevents the 'criticized' from listening and engaging a 'critic.' The opportunity to practice respectful and empathic responses to ensure clear understanding of the criticisms would be lost. Some complaints or critiques are appropriate and fair-minded (e.g., minimizing the miraculous in therapy, underuse of biblical wisdom in therapy) meriting individuals' attention and resolve to change.

Although responding to disparagements has been marked as a wise decision, there are some risks; confusion or discouragement may develop. About five years into the Monsignor's professorship, he became exhausted and depressed while engaging faultfinders and facing a possible termination. Pope Leo XIII's apostolic delegate, Satolli, engineered the removal of Bishop Keane, CUA's first president and recommended the dismissal of Pace (Gleason 1995; McAvoy 1957). In a draft version of his published article, Ryan (1961b) revealed: "In 1896, an attack on Pace's teaching [of scientific psychology] coincided with a period of (mental and) physical depression" (n.p.). Fortunately, the Pope did not act upon Satolli's recommendation for Dr. Pace's removal.

Another tactic is to share a personal explanation for involvement in the psychological field. Peoples' rationales will vary by purpose and content. One person's rejoinder cannot fit everyone or every situation; also, responses will most likely transform over time as experience in the discipline increases. To illustrate this application, we present our four-part rationale for working in the psychological field. First, we endeavor to become fluent or competent in psychology to honor God with our work, a passion for adaptive mental health, and to gain a place of influence among these scientists. It is a choice to be salt and light with non-religious psychological professionals, not avoid them (Matthew 5: 13–16). Second, we seek to glorify God by drawing out knowledge from the created order on human psychological functioning. It is a decision honoring the dominion command, not ignoring it (Genesis 1:28). Third, we are aware of the worldviews, non-theistic beliefs and practices, embraced by non-religious psychologists and disavow many of the tenets. It is a resolution to practice wisdom and discernment, not minimize the work of the Holy Spirit (Hebrews 5:14). Fourth, we intentionally nurture an enthusiastic relationship with God in an on-going manner to pursue holiness and intimacy with God while studying and applying psychological information. It is a determination to abide in Christ as "an apprentice of the Master's overall way of life" and not separate Him from our work (John 15:5) (Porter 2017, p. 2).

*5.4. Pace's Unrelenting Steadfastness to the Christian Faith—Nurture an Enthusiastic Attachment to God*

Pace's seminary experiences at PNAC and his experiences as a priest placed him in an ideal position to be anchored by a vibrant theological system. Yet, followers of Christ, past or present, chose/choose to nurture their faith. Sheehy (1938) testified about the enduring nature of the Monsignor's unrelenting steadfastness to God. He wrote:

In many visits paid to him during his residence in the hospital, I found such a profundity of faith and confidence in God . . . A remarkable life had been lived in such close communion with God that it gave his friends a feeling of his immortality. (Sheehy 1938, p. 4)

For psychologically-trained Protestants in the 21st century, choosing to nurture an enthusiastic attachment to God suggests an intentional, on-going journey of learning, processing, and applying biblical knowledge. The foci of such a quest are a dynamic foundation in Scripture (orthodoxy) and an authentic lifestyle of faith (orthoproxy) (Farnsworth 1985). The form and content of this journey will vary. Some Christians might engage the quest with a book study such as *Knowing God* (Packer 1993) or *Knowing Christ Today* (Willard 2009). For others, the nurturing process might start by clarifying what doctrines are believed and why.

When psychologically-trained Protestants engage non-religious psychological professionals, competency or fluency in key theological doctrines can be helpful. For instance, the naturalistic worldview of some psychologists does not include knowledge on the immanent, transcendent, and sovereign God of the universe. The mission of Christ and His life-changing teachings are also usually not recognized and metaphysical realities relative to human constitution, purpose, and destiny are often not considered. Therefore, clarity on theology proper, Christology, and biblical anthropology are vital.

We suggest any clarification of doctrines proceed beyond 'belief' and toward 'knowledge.' Distinguishing the terms, Willard (2009) argued the latter involves truth centered in appropriate evidence. The former is not bound to truth or evidence. Often, people believe "what is false" (p. 16). Interestingly, the modern American Protestant church tends to struggle in its approach to the teaching of doctrine. Willard (2014) maintained:

The proper role of doctrine is teaching openly with a view to people coming to understand things, not with a view to them winding up with the right views. The problem is that doctrine is [often] taught in a way that says you must believe this whether you believe or not. (p. 27)

## 6. Conclusions

Admired people can seem like heroes, even though, they are ordinary, imperfect individuals accomplishing extraordinary feats under exasperating circumstances. Associates closely connected with Edward A. Pace affirmed a remarkable set of personal characteristics underscoring the Bandurian traits of competency, prestige, and relevance. They described him as wise, prudent, shrewd, reserved, gentle, and bold (Brennan 1938; Ryan 1961a; The Tower 1938; Yearly 1938).

Speaking of Thomas Aquinas, the Monsignor believed:

Wise [people] are a power in the world, not alone for the knowledge they impart, but also and chiefly because their work invites others, encourages others, to imitate and perchance to surpass them. (Pace 1896, p. 188)

In similar fashion, the Monsignor's model, a set of habits—an appetence for wisdom in multiple disciplines, a keen awareness of rival worldviews, intentional ripostes to the Catholic critics of his era, and an unrelenting steadfastness to the Christian faith—invite psychologically-trained Protestants to imitate and surpass him.

This case study offered a glimpse into Dr. Pace's dedication, struggle, and influence while engaging at the intersection of Christianity/theology and the new psychology of the late 19th and early 20th centuries. It is illustrative of a person endeavoring toward whole-hearted commitment to the Christian faith and an admirable adherence to scientific psychology. Psychologically-trained Protestants in the 21st century can also possess faith-filled perseverance while participating in the cutting edge of scientific inquiry into psychological functioning (e.g., neuroscience). By emulating Pace in this manner, it is possible to participate in, rather than reactively respond to forces shaping the current zeitgeist of our time.

**Author Contributions:** Individual contributions include the following: Conceptualization, K.A.P.; writing—original draft preparation, K.A.P., K.G.P.; writing—review and editing, K.A.P., K.G.P.

**Funding:** This research received no external funding.

**Acknowledgments:** Archival research assistance resulted from the generous support of Shane T. MacDonald at the Catholic University of America in Washington D.C. and Christopher Gundlach at the University of Leipzig in Leipzig, Germany. Other assistants for manuscript support, conceptualization, and editorial assistance included John Drury, Dorothy Easterly, BJ Fratzke, and Jason Runyan.

**Conflicts of Interest:** The authors declare no conflict of interest.

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
