# Peer review of "Edward A. Pace: First-Generation Psychologist, Twenty-First Century Role Model"

_religions, doi:10.3390/rel10100590_

Round 1

Reviewer 1 Report

This is a well-researched essay and reads easily.  However, I would recommended cutting back on the numerous "emphasis added" to the citations. In many cases, the point made by the citation was quite clear without italics added to key words.  I also thought it would flow better if the application of Pace's habits to contemporary Protestants working in the field were all brought together after all the material on Pace had been presented.  

Reviewer 2 Report

This is an interesting article that I see having essentially three parts.  The first is a biographicakl sketch and study of Edward Pace a first generation psychologist and priest who studied with Wundt, made contributions to empirical psychology and unlike Brentano never became an apostate.  The authors use this fact to then explore a theological/philosophical defense of discernment that while never e4xplicity adavanced by Pace,  can be used by believing Protestants (these authors) to defend the involvement in psychology by Protestants who, like Pace, need not abandon their faith.

All this is very interesting but needs more critical development. I would recommend three issues be confronted.  First, the fact that Pace isolated phenomenon studies such that metaphysical issue could be avoided.  However,using William James (referenced in this article) this is unlikely to be possible.  Naturalism is at odds with Catholic and Protestant views in many areas and these are simply not addressed.  Second,. the appeal to scripture for discernment is a fair Protestant stance used by these authors but cannot be applied directly to Pace, Catholicism and the teaching magisterium of the church.  By these authors own admission, Pace was recommend for removal at his university. Third, the authors avoid any serious discussion of the direct conflicts between naturalism and biblical views endorsed by many Protestants and the simple appeal to discernement will not sp;ve the issues that need to be discussed.

Reviewer 3 Report

A well written manuscript that I enjoyed to read! Suitable for publication. The presentation of opposing takes on the soul captured nuances well too. James' struggled with the concept of the soul in his Principles of Psychology before the conclusion of the concept. Still I feel that he couldn't really let it go but produced alternate names like the judging thought, the I, the soul-substance etc. This you present nicely through the sayings of Pace (line 405-411) as he recounted: "Why doesn't the poor man say soul and be done with it." 

I only have two observations (the first is more important):

There is incongruity within your author and date citations. For example: Line 245 (Braun 1968), line 343 (Pace 1895a p. 144), line 435 (Misiak & Staudt 1954, p. 77) and line 481 (Hardin 2014, p. 324) in comparison to (for example) line 596 (Pace, 1895a, p. 149), line 574 (Pickern, 2000). Ergo: You use coma's in different ways so please control for a congruent author/date application.

You write with a specific interest to psychologically trained Protestant which is great. I also suggest that there may be more theological freedom and flexibility today (for psychologically trained protestants) than for example within a Catholic tradition... But on the other hand, the diversity may be greater within the Protestant family which may local differences more nuanced.

Best wishes!

Round 2

Reviewer 2 Report

The author has addressed adequately concernes with the original submission in this revision.